# Alpha_2_ Antagonist Vatinoxan Does Not Abolish the Preconditioning Effect of Dexmedetomidine on Experimental Ischaemia–Reperfusion Injury in the Equine Small Intestine

**DOI:** 10.3390/ani13172755

**Published:** 2023-08-30

**Authors:** Nicole Verhaar, Veronika Kopp, Christiane Pfarrer, Stephan Neudeck, Kathrin König, Karl Rohn, Sabine Kästner

**Affiliations:** 1Clinic for Horses, University of Veterinary Medicine Hannover, 30559 Hannover, Germany; 2Institute for Anatomy, University of Veterinary Medicine Hannover, 30559 Hannover, Germany; 3Small Animal Clinic, University of Veterinary Medicine Hannover, 30559 Hannover, Germany; 4Department of Biometry, University of Veterinary Medicine Hannover, 30559 Hannover, Germany

**Keywords:** horse, ischemic, jejunum, colic, anaesthesia, sedation, conditioning, MK467, vatinoxan

## Abstract

**Simple Summary:**

Intestinal disease with disruption of blood flow (ischaemia) is associated with a relatively high mortality rate in horses. Many investigations have focused on treatment strategies that reduce tissue injury. The sedative drug dexmedetomidine was found to precondition intestines subjected to ischaemia, resulting in reduced injury in different species, including horses. However, it remains unknown what exactly mediates this effect. Therefore, the aim of this study was to determine the effect of dexmedetomidine preconditioning with and without the administration of a drug called vatinoxan, which antagonizes the action of dexmedetomidine on the peripheral alpha_2_ adrenoreceptor. In 12 horses under general anaesthesia, intestinal ischaemia was implemented, followed by tissue reperfusion. Six horses received dexmedetomidine only (group Dex), while the other six horses received both dexmedetomidine and vatinoxan (DexV). Intestinal samples were taken to evaluate the degree of tissue injury, followed by a comparison between the groups. There was no difference before and directly after the disruption of tissue blood flow. After reperfusion, group DexV showed less mucosal injury compared to group Dex. There were no significant differences in programmed cell death between the treatment groups. In conclusion, antagonizing the peripheral alpha_2_ adrenoreceptors did not negatively affect dexmedetomidine preconditioning.

**Abstract:**

Pharmacological preconditioning with dexmedetomidine has been shown to ameliorate intestinal ischaemia reperfusion injury in different species, including horses. However, it remains unknown if this effect is related to alpha_2_ adrenoreceptor activity. Therefore, the aim of this study was to determine the effect of dexmedetomidine preconditioning with and without the administration of the peripheral alpha_2_ antagonist vatinoxan. This prospective randomized experimental trial included 12 horses equally divided between two treatment groups. Horses in group Dex received a bolus of dexmedetomidine followed by a continuous rate infusion (CRI), while group DexV additionally received vatinoxan as bolus and CRI. A median laparotomy was performed under general anaesthesia, and jejunal ischaemia was applied for 90 min, followed by 30 min of reperfusion. Mucosal damage was evaluated in full thickness biopsies by use of a semiquantitative mucosal injury score and by determining the apoptotic cell counts with immunohistochemical staining for cleaved caspase-3 and TUNEL. Comparisons between the groups and time points were performed using non-parametric tests (*p* < 0.05). During pre-ischaemia and ischaemia, no differences could be found in mucosal injury between the groups. After reperfusion, group DexV showed lower mucosal injury scores compared to group Dex. The apoptotic cell counts did not differ between the groups. In conclusion, antagonizing the peripheral alpha_2_ adrenoreceptors did not negatively affect dexmedetomidine preconditioning.

## 1. Introduction

Many studies have been performed to investigate treatment strategies that may ameliorate intestinal ischaemia reperfusion injury in horses. This includes the concept of preconditioning, which refers to the activation of intrinsic cell survival programs after exposure to mild ischaemic stimuli or pharmacologic agents [1]. Pharmacological preconditioning (PPC) with alpha_2_ agonists dexmedetomidine (Dex) and xylazine were shown to have mild protective effects in experimental jejunal ischaemia in horses, with lower mucosal damage scores as well as reduced apoptotic and inflammatory cell counts [2,3,4]. In laboratory animals, PPC with Dex also reduced intestinal injury [5,6,7]. Investigating the mechanism of action, several studies in laboratory animals found direct inhibition of apoptosis and inflammation, yet the exact mechanism of activation remains unclear [7,8,9].

Dex produces its sedative effects via the activation of central alpha_2_ adrenoreceptors [10], while it induces cardiovascular effects such as vasoconstriction through activation of the peripheral alpha_2_ adrenoreceptors [11] but also via interaction with imidazoline receptors [12]. One author has reported that the addition of alpha_2_ antagonist yohimbine abolished the intestinal protective effect of Dex in rats, leading to the disappearance of its anti-apoptotic and anti-inflammatory effect [13]. This suggests that the protective action of Dex may be dependent on the activation of the alpha_2_ adrenoreceptor, possibly by mimicking ischaemic preconditioning through vasoconstriction.

Vatinoxan (MK-467) is a peripheral alpha_2_ adrenoceptor antagonist that hardly crosses the blood/brain barrier due to its poor lipid solubility [14]. It has been shown to ameliorate alpha_2_ agonist-related side effects, such as bradycardia and reduced cardiac output in horses [15,16,17]. When used as premedication, it can induce hypotension during general anaesthesia, requiring increased cardiovascular support to maintain normotension [18,19].

To the authors’ knowledge, there are no publications investigating peripheral alpha_2_ adrenergic antagonists during Dex PPC. The aim of this study was to determine if the protective effect of dexmedetomidine in horses is related to peripheral alpha_2_ receptor agonism. The authors hypothesized that following treatment with Dex PPC, the antagonization of peripheral alpha_2_ adrenoreceptors with vatinoxan would worsen intestinal mucosal damage compared to PPC with Dex alone.

## 2. Materials and Methods

### 2.1. Animals

This study was reviewed by the Ethics Committee for Animal Experiments of Lower Saxony, Germany, and approved according to the German Animal Welfare Act (LAVES 33.19-42502-04-16/2212). For this prospective terminal in vivo experimental trial, 12 horses owned by the university were divided into 2 groups of 6 horses each. This allocation was conducted by simple randomization with an equal allocation ratio. The first group, consisting of warmbloods only, was subjected to preconditioning with dexmedetomidine (group Dex) and had a mean age of 8.5 y and weight of 518 kg. The second group was also preconditioned with dexmedetomidine, but together with peripheral alpha_2_ adrenoreceptor antagonist vatinoxan (group DexV), and consisted of 4 warmbloods, one islandic horse, and one trotter with a mean age and weight of 11.2 y and 507 kg, respectively. At least two weeks prior to surgery, they were stabled in the equine clinic. The horses were elected for euthanasia due to problems unrelated to the gastrointestinal or cardiovascular system, such as orthopaedic disease. All were systemically healthy without any signs of gastrointestinal or cardiovascular disorders.

### 2.2. Anaesthetic and Surgical Protocol

First, all horses in both groups received 3.5 µg/kg dexmedetomidine (Dexdomitor, 0.5 mg/mL, Orion Pharma GmbH, Hamburg, Germany) IV as premedication, immediately followed by the initiation of a continuous rate infusion (CRI) of 7 µg/kg/h that was continued throughout the complete procedure. Group DexV additionally received a bolus of 130 µg/kg vatinoxan (Vetcare Ltd., Helsinki, Finland) IV as well as a CRI of vatinoxan at a rate of 40 µg/kg/h. Dexmedetomidine and vatinoxan were infused concurrently. Anaesthesia was induced with 0.05 mg/kg diazepam (Ziapam 5 mg/mL, Ecuphar GmbH, Greifswald, Germany), and 2.2 mg/kg ketamine (Narketan 100 mg/mL, Vetoquinol GmbH, Ismaning, Germany). Following endotracheal intubation, anaesthesia was maintained with Isoflurane (Isofluran CP, CP-Pharma GmbH, Burgdorf, Germany) in 100% O_2_ at an end-tidal pressure of 1.1 vol% with intermittent positive pressure ventilation with a positive inspiratory pressure of 20–25 cm H_2_O. The respiratory rate was adjusted to maintain normocapnia of an end-tidal carbon dioxide partial pressure of 35–45 mmHg. Lactated Ringer’s (Ringer-Laktat 5 L, B. Braun SE, Melsungen, Germany) solution was administered at 5 mL/kg/h. In cases of a decrease in the mean arterial pressure below 60 mmHg, the infusion rate was increased to 10 mL/kg/h, together with a dobutamine CRI (Dobutamin-ratiopharm 250 mg, Ratiopharm GmbH, Ulm, Germany) of 0.33 μg/kg/min to effect.

The horses were positioned in dorsal recumbency, and a median laparotomy was performed sixty minutes after induction of anaesthesia. Thirty minutes later, experimental low-flow ischaemia was induced in the distal jejunum as described previously [2]. In brief, the mesentery and intestine were included in a mass ligature with umbilical tape that was tightened under monitoring of tissue blood flow and saturation by micro-lightguide spectrophotometry and laser Doppler flowmetry (O2C Oxygen to See, Lea Medizintechnik GmbH, Gießen, Germany) [20] until the blood flow was reduced to 10% of the measurement before ligation. Ninety minutes after initiation of ischaemia, the ligature was released. Following 30 min of reperfusion, the horses were euthanized by intravenous administration of 90 mg/kg pentobarbital without regaining consciousness.

The perfusion and oxygenation data, as well as the plasma concentrations of dexmedetomidine and vatinoxan, have previously been published [21].

### 2.3. Histology

Full-thickness intestinal samples were taken immediately prior to ischaemia (pre-ischaemia sample, P). Following 90 min of ischaemia before release of the ligature (ischaemia sample, I) and after 30 min of reperfusion (reperfusion sample, R), full thickness samples were taken from the center of the (post-)ischaemic segment. The tissue was fixed in 4% formaldehyde and embedded in paraffin. The slides were stained routinely with haematoxylin and eosin for histomorphological examination of the mucosa using a modified Chiu score (Table 1) [22,23]. This was assessed by light microscopy (AXIO Scope.A1, Carl Zeiss Vision GmbH, Aalen, Germany) in 10 adjoined high-power fields (hpfs) at a 400-fold magnification. The observer was blinded to the identity of the slides. Each field of view was scored individually, and they were averaged. Immunohistochemical staining was performed for cleaved caspase-3 (casp3) (CleavedCaspase-3Asp175 antibody, Cell Signaling Technology Europe B.V., Leiden, The Netherlands) as a marker for apoptosis, as described previously [2]. Furthermore, terminal deoxynucleotidyl transferase dUTP nick-end labeling (TUNEL) (ApopTag^®^, Merck KGaA, Darmstadt, Germany) was used as a marker for late apoptosis and early cell necrosis, as described previously [2]. The cells in the mucosa showing clear immunoreactivity were counted in 10 adjoined hpfs per slide, using exact surface measurements (Axiocam 105 color and Software ZEN 2.3, Carl Zeiss Vison GmbH) to determine the number of stained cells per mm^2^.

### 2.4. Statistical Analyses

Prior to commencing this study, a power analysis was performed to calculate the necessary sample size (G*Power 3.1.9.2, Heinrich Heine Universität, Düsseldorf, Germany). To detect a difference between the two treatment groups of one grade in the histomorphology score with a standard deviation of 0.5, based on a power of 0.8 and an alpha of 0.05, a sample size of 6 horses per group was required.

Statistical analysis and graph design were performed with commercial software (SAS 9.4m5 with the Enterprise Guide Client 7.15, SAS Institute Inc., Cary, NC, USA; GraphpadPrism 9.4.1, Graphpad Software Inc., San Diego, CA, USA). A *p*-value of <0.05 was considered significant. Testing for normal distribution was conducted by visual assessment of the q–q plots of the model residuals and the Shapiro–Wilk test. The data were neither normal- nor lognormal distributed and were expressed as the median (min–max). The comparison between the groups and time points was performed with distribution-free nonparametric models. For comparing the correlated different time points, a permutation test (an exact Friedman test) was used, with a post hoc Sidak test for multiple pairwise comparisons (SAS macro RIBDPERM.MAC, Institut für Angewandte Mathematik und Statistik, Universität Hohenheim, Stuttgart, Germany). A Wilcoxon two-sample test was used to compare the results between the different groups at each time point.

## 3. Results

During pre-ischaemia, none of the samples had any histomorphological signs of mucosal damage, with all samples showing a mucosal injury score of 0. Significant damage occurred during ischaemia, with significant increase in mucosal injury in both groups (group Dex *p* = 0.004; group DexV *p* = 0.007) (Figure 1). There was no significant difference between ischaemia and reperfusion. Comparing the treatment groups, the mucosal injury did not differ significantly between the groups at the time points of pre-ischaemia (*p* = 1) and ischaemia (*p* = 0.12). However, during reperfusion, group DexV had a lower mucosal injury score than group Dex (*p* = 0.03).

Evaluating the casp3 positive cell counts resulted in around five apoptotic cells per mm^2^, which did not increase significantly during ischaemia and reperfusion (Figure 1). Furthermore, there were no significant differences between the groups.

During pre-ischaemia, both groups showed around five TUNEL positive cells per mm^2^, comparable to the casp3 positive cell count (Figure 1). In group DexV, the cell count increased significantly during ischaemia (*p* = 0.02), and it decreased significantly in group Dex during reperfusion (*p* = 0.03), without significant differences in the group comparison.

## 4. Discussion

This study describes the progression of ischaemia reperfusion injury following pharmacological preconditioning with dexmedetomidine with and without the addition of the peripheral alpha_2_ antagonist vatinoxan. The main finding was that the horses receiving Dex together with vatinoxan did not show worsening of the ischaemic injury, rejecting this hypothesis. Moreover, the group receiving both dexmedetomidine and vatinoxan showed less mucosal damage during reperfusion than the ones treated with Dex only. Therefore, we rejected our hypothesis that the antagonization of peripheral alpha_2_ adrenoreceptors would worsen intestinal mucosal damage in comparison to PPC with Dex alone.

The results of the current study could indicate that the peripheral alpha_2_ adrenoceptors are not key to the preconditioning effect of Dex, suggesting another path of activation. We cannot be sure that the dose of vatinoxan has antagonized the complete Dex effect at the receptor level; however, the perfusion and oxygenation data have shown that the peripheral effects were effectively antagonized [21]. Furthermore, the dose used in the current study was based on previous reports with effective use of vatinoxan antagonizing 7 µg/kg medetomidine [18]. Contrary to our results, a previous study investigating intestinal PPC in rats reported that the coadministration of yohimbine as general alpha_2_ antagonist abolished the protective effect of Dex [13]. The discrepancy between the two studies could be caused by differences in experimental set-up or the species used. An additional difference lies in the alpha_2_: alpha_1_ binding ratio between vatinoxan (105:1) and yohimbine (40:1), with yohimbine having more affinity to alpha_1_ adrenoreceptors than vatinoxan [14]. Another option could be that Dex PPC involves central alpha_2_ adrenoreceptors that are not inhibited by vatinoxan [10,14,17], knowing that Dex-induced stimulation of these central receptors can enhance vagal neural activity, which may elicit a protective response [24].

The cardiovascular effects of dexmedetomidine need to be taken into account, considering that dexmedetomidine has been reported to both negatively and positively affect intestinal perfusion [19,25,26]. Of the horses included in the current study, the perfusion and oxygenation data as well as the plasma concentrations of dexmedetomidine and vatinoxan have already been published [21]. Relevant for the interpretation of the concurrent intestinal ischaemia reperfusion injury is that the horses treated with vatinoxan had superior oxygenation variables and tissue perfusion, which was also shown in previous studies [17,19,27]. Furthermore, more horses in group DexV needed a dobutamine CRI to maintain normotension [28]. The improved tissue perfusion in group DexV could be a possible explanation for the lower mucosal injury score in the DexV group during reperfusion.

In the field of cardiological ischaemic preconditioning, the alpha_1_ adrenoceptor has been suggested to play a role in the activation of the protective effect in rats and rabbits [29,30]. This theory is not very plausible for Dex PPC, considering the selectivity of this drug for the alpha_2_ adrenoceptor. Moreover, another study in horses found that the less selective alpha_2_ adrenoceptor agonist xylazine did not influence the mucosal injury score but only reduced apoptotic cell counts [2]. Even though a direct comparison is not available, this may be interpreted as xylazine being less protective than Dex [3,4], possibly indicating the importance of the selective alpha_2_ adrenoceptor agonistic effect for the mechanism of PPC. Alternatively, the action of Dex on imidazoline receptors [12] may play a role in the PPC, considering vatinoxan is a non-imidazoline alpha_2_ adrenergic antagonist. However, this receptor has not been investigated in the PPC of the intestine.

The main limitation of this study is the lack of an untreated control group. We elected to use the information from a control group in previously published work by this research group following the identical protocol [3] to verify the protective effect of Dex PPC. This was preferred over the inclusion of another group in this study, as this would have called for at least six additional horses in this terminal experiment. Looking at the mucosal damage of the Dex-treated groups in both studies, the mucosal injury grades and apoptotic cell counts are very much alike, indicating a similar effect of Dex in the current study. Furthermore, the current study mainly focuses on the difference with and without vatinoxan, building on the fact that several other publications have already shown dexmedetomidine’s protective potential. Another important limitation is the short duration of reperfusion that limits the investigation of the treatment effect beyond this time frame. Furthermore, the anaesthetic maintenance with isoflurane could have elicited a preconditioning effect of its own [31,32]; however, this would have been present in both groups.

Previous studies have found mild protective effects of PPC with Dex, yet it cannot be expected to reduce mucosal injury during intestinal ischaemia in horses completely. Therefore, it should only be seen as an adjunctive therapy, with application in horses that are not eligible for surgical treatment or during anaesthetic management of surgical cases as possible indications. It is obvious that the clinical application of this technique in colic patients is limited because the occurrence of a strangulating lesion cannot be predicted. However, it may have potential to ameliorate injury in cases where additional intestinal segments are progressively incorporated or where the strangulated intestine still receives some blood flow and is not maximally injured at the time of intervention.

## 5. Conclusions

The peripheral alpha_2_ antagonist vatinoxan did not abolish the preconditioning effect of dexmedetomidine on experimental ischaemia–reperfusion injury in the equine small intestine, indicating that Dex PPC is not mediated solely through activation of peripheral alpha_2_ adrenoreceptors. Moreover, less mucosal injury was seen in the group that received both dexmedetomidine and vatinoxan. This suggests that antagonization of the peripheral side effects of Dex, such as intestinal hypoperfusion or hypomotility, may be performed without negatively affecting the protective potential. However, it must be noted that the improved intestinal perfusion observed with vatinoxan administration cannot be ruled out as possible cause for the decreased tissue injury. The clarification of the Dex PPC mechanism of action in tissue cultures or organoids is indispensable, as this may allow the use of this treatment strategy in clinical patients and optimize its treatment potential.

## Figures and Tables

**Figure 1 animals-13-02755-f001:**
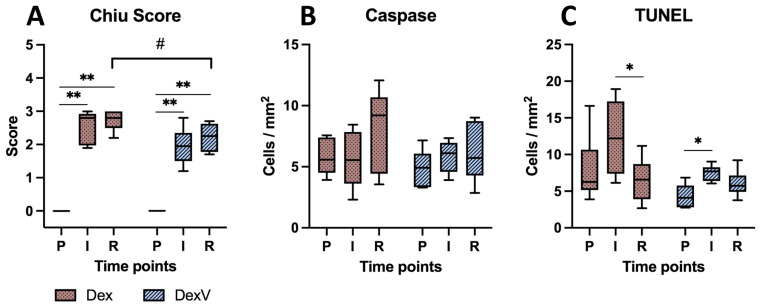
Box-plot diagram depicting the modified Chiu score for intestinal mucosal histomorphological damage (**A**), cleaved caspase-3 positive cells (**B**), and terminal deoxynucleotidyl transferase dUTP nick-end labelling positive cells (**C**) in horses subjected to experimental jejunal ischaemia and pharmacological preconditioning with dexmedetomidine (group Dex) or the coadministration of dexmedetomidine and vatinoxan (group DexV). Significant differences between different time points within the groups are marked with an asterisk (* *p* < 0.05; ** *p* < 0.01), significant differences between the groups with a hashtag (# *p* < 0.05). P = Pre-ischaemia; I = Ischaemia; R = Reperfusion. The diagram displays the median as a horizontal bar, the interquartile range as the box, and the minimum and maximum as the whiskers.

**Table 1 animals-13-02755-t001:** Modified Chiu score [22] for small intestinal mucosal histomorphology.

Grade	Description
0	Normal mucosal villi
1	Development of subepithelial (Gruenhagen’s) space at the apex of the villus
2	Extension of the subepithelial space with moderate lifting of the lamina propria
3	Significant epithelial separation down the villus sides, until halfway down the villus
4	Denuded villi with the lamina propria exposed
5	Digestion and disintegration of the lamina propria and ulceration

## Data Availability

The data that support the findings of this study are openly available under the following reference: Verhaar, Nicole (2023), “Effect of Vatinoxan on dexmedetomidine preconditioning on experimental ischaemia–reperfusion injury in the equine small intestine”, Mendeley Data, V1, https://doi.org/10.17632/cfv4v5c7ct.1.

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
