# Peer review of "Alpha2 Antagonist Vatinoxan Does Not Abolish the Preconditioning Effect of Dexmedetomidine on Experimental Ischaemia–Reperfusion Injury in the Equine Small Intestine"

_animals, 2023, doi:10.3390/ani13172755_

Round 1

Reviewer 1 Report

Interesting report.

The manuscript reports that the addition of vatinoxan did not alter the preconditioning effect of dexmedetomidine administered prior to the induction of ischemia in the equine small intestine.  The report details a portion of a larger study on the effects of the drugs. 

Interestingly, the authors report less mucosal injury when both vatinoxan and dexmedetomidine were administered. In the discussion, the authors note that more horses in the vatinoxan/dexmedetomidine group required dobutamine to maintain arterial blood pressure at the desired level during anesthesia. The reviewer questions whether the revelation that more horses in the vatinoxan/dexmedetomidine group required dobutamine should appear in the results rather than in the discussion.  Further, should the incidence of dobutamine use and the dose of dobutamine correlate to the presence of or the degree of mucosal injury?   If they have examined this, it should be so stated and their thoughts should be presented.  

The reviewer recognizes that the authors do not necessarily want to publish data in two places, but consideration of the effects of dobutamine and the potential effects on blood flow are germane to the consideration of blood flow.

Another point of inquiry is whether or not the individuals performing the histological analysis were blinded to the treatment group and the point in the treatment when the sample was obtained (pre-ischemia, ischemia, reperfusion)?  It would seem appropriate.

Author Response

Dear reviewer,

Thank you very much for taking the time to review our manuscript. We would like to thank you for your comment, which we have addressed in the point-by-point response below:

  • The reviewer questions whether the revelation that more horses in the vatinoxan/dexmedetomidine group required dobutamine should appear in the results rather than in the discussion. Further, should the incidence of dobutamine use and the dose of dobutamine correlate to the presence of or the degree of mucosal injury?   If they have examined this, it should be so stated and their thoughts should be presented.  The reviewer recognizes that the authors do not necessarily want to publish data in two places, but consideration of the effects of dobutamine and the potential effects on blood flow are germane to the consideration of blood flow.
    • We fully agree with this point that the cardiovascular status of the horses is of relevance for the interpretation of the data. We have added the reference to the cardiovascular results to the materials and methods section (Line 125-126), so that the reader is made aware of these results earlier. However, we do not believe that the results of another published paper can be mentioned in the results of a new paper. We did not perform a correlation analysis of the histology scores and the dobutamine doses. We believe that this would be statistically tricky and unlikely to yield a meaningful result, considering the low sample size and individual variation between the horses.
  • Another point of inquiry is whether or not the individuals performing the histological analysis were blinded to the treatment group and the point in the treatment when the sample was obtained (pre-ischemia, ischemia, reperfusion)?  It would seem appropriate.
    • Yes, the observer was blinded for the identity of the slides. This was added to the text.(Line 136)

Reviewer 2 Report

This is an interesting study about the effects of vatinoxan on the preconditioning effect of dexmedetomidine. Six horses were included in each group, dexmedetomidine and dexmedetomidine vatinoxan. Samples at three times were taken and analysed. Vatinoxan did not abolish the preconditioning effect of dexmedetomidine, even improved it. It is unknown how dexmedetomidine does this preconditioning effect and it is hypothesised the imidazole receptors might play a role in it.

Keywords: “vatinoxan” might be included here.

Introduction: I miss a couple of sentences explaining how vatinoxan works and the current information about its effects in horses. Vatinoxan has shown to have cardiovascular effects during anesthesia with isoflurane in horses.

Material and methods:

Line 93: How long did it pass between dexmedetomidine bolus and CRI? When were vatinoxan bolus and CRI given?

Line 101: correct “H2O” (subscript).

Line 103: correct “5 ml kg/kg/h”. I think it means 5 ml/kg/h.

Line 123: “H&E” are not defined previously as hematoxylin and eosin stain.

Why 30 minutes of reperfusion were study and not a longer period?

Results:

What about blood pressure? Were differences between groups? In how many horses were fluids increased to 10 ml/kg/h? Use of dobutamine? Doses? I think this is important when discussing ischaemia and repercussion damages. It is needed to be included. I also think PaCO2, PaO2 and pH should be considered.

Did you need any top-up bolus of dexmedetomidine during sedation or ketamine during induction or maintenance?

Figure 1: IQR: interquartile range? It is not defined.

Line 161: “around” sounds inexact and no scientific   

Discussion

First paragraph: your hypothesis here is unnecessary.

Line 191: where are reported the  oxygenation data?

Line 197: which specie was used in the other study?

Line 214: which drugs and species exactly?

Cardiovascular differences between groups should be discussed.

Conclusion

There is no mention about better results in dex-vatinoxan group than in dex group, it was mentioned in the first paragraph of the discussion. Or was it wrong?

Thank you very much for this interesting work.

Author Response

Dear reviewer,

Thank you very much for your thorough review and the constructive comments. We have addressed your comments in the following point-by-point response:

  • Keywords: “vatinoxan” might be included here.
    • This was added to the keywords. (Line 48)
  • Introduction: I miss a couple of sentences explaining how vatinoxan works and the current information about its effects in horses. Vatinoxan has shown to have cardiovascular effects during anesthesia with isoflurane in horses.
    • We added this to the introduction. (Line 69 – 74)
  • Material and methods: Line 93: How long did it pass between dexmedetomidine bolus and CRI? When were vatinoxan bolus and CRI given?
    • This information was added to the main text. (Line 102-103)
  • Line 101: correct “H2O” (subscript).
    • Thank you for pointing this typo out to us. This was changed in the text. (line 108)
  • Line 103: correct “5 ml kg/kg/h”. I think it means 5 ml/kg/h.
    • Thank you for pointing this typo out to us. This was changed in the text. (line 111)
  • Line 123: “H&E” are not defined previously as hematoxylin and eosin stain.
    • We added this information to the main text. (Line 133)
  • Why 30 minutes of reperfusion were study and not a longer period?
    • The reason for this was the long total time of anaesthesia required for this experiment, and the possibility of cardiovascular deterioration of horses during prolongued genereal anaesthesia. Therefore, we wanted to limit the total time of anaesthesia as much as possible, and pre-ischaemia and ischaemia being judged as the most significant time frame for this type of conditioning, resulting in a relatively short reperfusion time. If the reviewer wishes, we can add this to the discussion of the paper.
  • Results:What about blood pressure? Were differences between groups? In how many horses were fluids increased to 10 ml/kg/h? Use of dobutamine? Doses? I think this is important when discussing ischaemia and repercussion damages. It is needed to be included. I also think PaCO2, PaO2 and pH should be considered.
    • We fully agree with this point that the cardiovascular status of the horses is of relevance for the interpretation of the data. We have added the reference to the cardiovascular results to the materials and methods section, so that the reader is made aware of these results earlier (Line 125-126). However, we do not believe that the results of another published paper can be mentioned in the results of a new paper.
  • Did you need any top-up bolus of dexmedetomidine during sedation or ketamine during induction or maintenance?
    • The described dose of Dex was set, and no additional top-ups were given. An additional ketamine bolus was given following induction only if ET-intubation or moving the horse on the surgery table was not possible due to inadequate anaesthesia depth.
  • Figure 1: IQR: interquartile range? It is not defined.
    • We have added this information to the text. (Line 188)
  • Line 161: “around” sounds inexact and no scientific
    • This was removed from the text.
  • Discussion - First paragraph: your hypothesis here is unnecessary.
    • We changed the wording of this section. (Line 197-198)
  • Line 191: where are reported the oxygenation data?
    • See our remark above, we added the reference here again too. (Line 203)
  • Line 197: which specie was used in the other study?
    • Rats – this was added to the main text. (Line 206)
  • Line 214: which drugs and species exactly?
    • This added to the main text. (Line 226-227)
  • Cardiovascular differences between groups should be discussed.
    • This was discussed in the third paragraph of the discussion.
  • Conclusion: There is no mention about better results in dex-vatinoxan group than in dex group, it was mentioned in the first paragraph of the discussion. Or was it wrong?
    • It was correct that we found that. We added it to the conclusion. (Line 263 – 268)

Reviewer 3 Report

In this prospective randomized paper the authors aimed to clarify the mechanism used by Dexmedetomidine to produce a pharmacological preconditioning (PPC) in horses that received an experimental intestinal ischemia. they hypothesis was that the alpha 2 adrenoreceptor antagonist Vatinoxan would abolish the PPC of dex. If the hypothesis would have been validated, they could conclude that dex PPC was procured via the activation of this receptor.

the authors provided a good description of material and method and results. 

Although well written I have a couple of question and suggestions:

- reference n.1-2: check if the references are in agreement with the paper requirements.

- when was vatinoxan administered ? together with dex? if after, how long afterwards ?

- please clarify how subject were allocated to each group? 

- I suppose that an owner's consent was used. please add a sentence about this

- I would suggest to add values of cardiovascular parameters in the group and assess for differences- three hours and 30 min anesthesia are a long time. cardiovascular differences among groups could have been responsible of the results.

Author Response

Dear reviewer,

Thank you very much for reviewing our manuscript and for the constructive comments. We have addressed your comments in the following point-by-point response:

  • reference n.1-2: check if the references are in agreement with the paper requirements.
    • I am not completely sure what you mean, but I will explain the reasons for including these references: reference 1 is an extensive review work/collection on different practices of conditioning and therefore in our opinion it fits the purpose. Reference 2 is the only work investigating pharmacological conditioning with xylazine in horses so far, and therefore should be mentioned here.
  • when was vatinoxan administered ? together with dex? if after, how long afterwards ?
    • This was administered at the same time. This information was added to the text. (Line 102-103)
  • please clarify how subject were allocated to each group? 
    • By simple randomisation with equal allocation ratio. This was added to the text. (Line 86-87)
  • I suppose that an owner's consent was used. please add a sentence about this
    • The horses were owned by the university. We added this to the text. (Line 86)
  • I would suggest to add values of cardiovascular parameters in the group and assess for differences- three hours and 30 min anesthesia are a long time. cardiovascular differences among groups could have been responsible of the results.
    • We fully agree with this point that the cardiovascular status of the horses is of relevance for the interpretation of the data. These results have been published previously, and therefore this is not included in the results section of the current manuscript. We have added a reference to this paper in the materials&methods sections, so that the reader knows of this prior to reading the results (Line 125-126).

Round 2

Reviewer 2 Report

Thank you very much for your answers. Congratulations for the paper. 

Line 107: "O2" subscript